# AI and Biotechnology to Combat Aflatoxins: Future Directions for Modern Technologies in Reducing Aflatoxin Risk

**DOI:** 10.3390/toxins17110524

**Published:** 2025-10-23

**Authors:** Charitha J. Gamlath, Felicia Wu

**Affiliations:** 1Department of Food Science and Human Nutrition, Michigan State University, East Lansing, MI 48824, USA; pahalaga@msu.edu; 2Department of Agricultural, Food, and Resource Economics, Michigan State University, East Lansing, MI 48824, USA

**Keywords:** AI, artificial intelligence, biotechnology, aflatoxins, control strategies

## Abstract

Although a decades-old problem in food safety, aflatoxin has largely resisted human control methods. This situation could be mitigated using new technologies that could provide better control all along the food supply chain, for crops frequently infected with the causative fungi *Aspergillus flavus* and *A. parasiticus*, which produce aflatoxin. Generative artificial intelligence (AI) and modern biotechnology could offer, and have offered, a suite of potential solutions to reducing both fungal infection and aflatoxin contamination of foods. In this paper, we describe these technologies, as well as means by which they may be utilized to reduce aflatoxin risk along the food supply chain. We discuss how regulatory frameworks worldwide may be restrictive for biotechnologies in certain parts of the world, but are relatively less stringent for AI at present. To the extent that these technologies can be harnessed and deployed safely to combat the problem of aflatoxins, we encourage research and development in these areas to improve the precision, accuracy, and speed by which to deal with this food safety risk.

## 1. Introduction

This article describes the potential applications of a convergence of technologies, artificial intelligence (AI) and biotechnologies, to reduce aflatoxin contamination in food and feed. Artificial intelligence (AI) has existed as a branch of computer science and an application in technology for decades. But in recent years, AI has garnered greater attention across multiple fields because more advanced forms have become much more accessible to the general public for the first time. Moreover, one does not need to know the details of how AI works to be able to use it across many domains. AI tools such as ChatGPT-5 or Gemini (Google AI) allow users to do many things: gather summary information on any topic they seek, do both rudimentary and complex calculations, write programming code or poems, create images, rewrite documents in particular styles, formats, and languages, detect patterns in data and images, design spaces, help with household and work tasks, and much more.

On a larger scale, AI is being used in industries and governmental and non-governmental organizations, in diverse ways. One of the ways that the use of AI could be enhanced in agriculture, food, and public health is in conjunction with emerging biotechnologies that help to curb risks associated with climate change, economic shocks, and geopolitical instability. Unlike AI, biotechnology has been used in agriculture since the 1980s, for multiple purposes: to prevent the freezing of citrus fruits (ice-minus: genetically engineered bacteria), to confer viral or insect pest resistance or herbicide tolerance in crops, to biofortify grains with micronutrients, to enhance growth rates of salmon, and to produce textiles from crops. One paper describes how biotechnology can be used to reduce mycotoxins in corn, either by reducing fungal infection or by degrading the toxins [1]. Combined with AI, biotechnologies could become more selective and informed in mitigating aflatoxin risk.

In this paper, we explore how AI and modern biotechnologies can be harnessed to advance aflatoxin-related research, detection, and mitigation strategies from farm to fork. We begin by providing an overview of aflatoxins, AI, and contemporary biotechnological tools. Subsequently, we synthesize the existing literature to highlight potential applications and their benefits for aflatoxin control and the promotion of human health. Finally, we propose a conceptual scenario to illustrate how AI and biotechnology can be incorporated into the entire food supply chain to manage aflatoxin contamination, using corn—one of the most susceptible crops—as a case study. This work offers a unique perspective on leveraging AI and modern biotechnological innovations for comprehensive aflatoxin management, aiming to inform future research, identify existing knowledge gaps, and accelerate the effective integration of emerging technologies in this field.

## 2. Background: Aflatoxins

Aflatoxins are toxic and carcinogenic secondary metabolites of the fungi *Aspergillus flavus* and *A. parasiticus*. As these fungi infect the food and feed crops such as corn, peanuts, tree nuts, cottonseed, and sunflower seeds grown in tropical and subtropical climates, aflatoxin is frequently a contaminant of these foods. Aflatoxins were first discovered in 1960 in a gruesome manner: when over 100,000 turkey poults died in the United Kingdom as a result of eating moldy peanut meal [2]. Since then, the exposure of aflatoxins was extensively investigated and subsequently recognized to cause liver toxicity and cancer (hepatocellular carcinoma) in multiple animal species. In 1993, the International Agency for Research on Cancer (IARC) classified “naturally occurring mixtures of aflatoxins” as a Group 1 carcinogen [3]. Further, aflatoxins are particularly carcinogenic in individuals exposed to high levels of both aflatoxins and hepatitis B virus (HBV). Exposure to both agents increases lifetime liver cancer risk multiplicatively when compared with the risk imposed by just one agent [4,5,6].

Out of the four common aflatoxins (B1, B2, G1, and G2), aflatoxin B1 (AFB1) is of the greatest concern worldwide, due to its toxicity, carcinogenicity, and prevalence in staple foods and feeds worldwide [7]. When ingested, AFB1 is metabolized in the liver by the P450 enzyme system into the carcinogenic intermediate, aflatoxin B1-8,9-epoxide (AFBO). This compound exists in two isomeric forms: endo-8,9-epoxide and exo-8,9-epoxide. Due to its highly electrophilic nature, AFBO readily reacts with biological amines in proteins and nucleic acids. When interacting with DNA, AFBO forms a covalent bond at the N7 position of guanine, producing the adduct AFB1-N7-guanine. The AFBO exo isomer exhibits a much greater affinity for guanine residues than the endo isomer and is therefore regarded as the primary carcinogenic metabolite [8]. The resulting DNA damage can initiate the development of hepatocellular carcinoma (liver cancer), or, at high exposure levels, cause acute liver failure leading to aflatoxicosis [9]. Increasing evidence also suggest that AFB1 can induce oxidative stress in cell function [10]. Globally, 5–28% of all liver cancer cases can be attributed to aflatoxin exposure, exceeding over 100,000 new aflatoxin-induced liver cancer cases per year. The majority of these cases occur in sub-Saharan Africa, Southeast Asia, and China, where both HBV infection rates and aflatoxin contamination in food remain high and poorly controlled [11]. To minimize dietary exposure to AFB1, regulatory limits have been established globally to ensure that contamination levels in food and feed remain sufficiently low. Regulatory limits for AFB1 in maize and peanuts—two of the most frequently contaminated food commodities, set by various nations and regulatory authorities are available in Wu et al. (2013) [12].

While farm-to-fork strategies currently exist to reduce aflatoxin in food, or its harmful effects in the body [13], oftentimes these strategies can be difficult to implement in real time, in the populations of interest. For example, biocontrol—atoxigenic strains of *Aspergillus* applied to crop soils to competitively exclude toxigenic strains on crops—can only be applied at the start of the growing season, before a farmer knows whether aflatoxin would even be a problem in that season. If later in the summer, the farmer detects high aflatoxin risk in the crops, it is too late then to apply biocontrol [14]. Similarly: many dietary chemopreventive strategies, such as incorporation of *Brassica* or *Allium* vegetables to the diet to reduce the risk of aflatoxin epoxide-mediated DNA damage [15], are not always feasible, as they are not readily available in sufficient amounts in the diets of populations where aflatoxin-contaminated maize and peanut consumption is high; and public awareness of these dietary chemopreventive agents is low.

The combined use of AI and modern biotechnologies could enhance the utility of these existing aflatoxin control strategies, as well as provide new strategies. These technologies are described below.

## 3. Background: AI

The term artificial intelligence (AI) refers to a suite of tools that simulates human intelligence and ability to perform tasks by means of machines; principally computer systems, robotics, and digital equipment [16,17]. Specifically, generative AI refers to AI that creates new material by processing large amounts of existing data. How can it do this? *Machine learning* (ML) is one of the key themes of AI. It involves the development of algorithms or models, sometimes called large language models or LLM, that are trained from ideally copious amounts of data to identify patterns or make predictions [18]. It is by machine learning that AI tools such as ChatGPT and Google Gemini can provide answers to the questions that users ask on virtually any topic: solving mathematical problems, helping to write essays and letters, coding, providing interior decorating or grocery shopping tips, or even imitating a therapist. This paper will not delve into the ethical and moral hazards associated with AI use, but will focus very specifically on possible applications in aflatoxin reduction.

Several statistical and mathematical methods are used when developing such algorithms. Some of the commonly used ML algorithms used in Aflatoxin control are described below:

*Artificial neural networks (ANN)* are a computational method that mimics the biological neural functions of the human brain. ANN consists of layers that are interdependent of each other. Each layer consists of processing nodes or “neurons,” linked by weighted connections that transfers outputs from one nodes input to another [19]. When the weighted sum of inputs exceeds a predefined threshold value of a node, it passes the signal to neighboring nodes via a function known as the transfer function [20]. Neural networks have many applications such as pattern recognition, prediction, and classifications, and are employed in the detection of fungal growth by image processing and for identifying complex metabolic pathways that lead to toxin-induced cellular alteration using molecular descriptors [19,21,22]. 

*Principal component analysis (PCA)* is a dimension reduction method that has commonly been used in research long before the advent of generative AI. PCA is useful when analyzing highly correlated data with a large number of variables, with multiple observations associated with each variable. PCA identifies key categories (referred to as principal components) that represent the original data with minimal loss of information. It projects the data into a new-lower dimensional subspace and reduces storage space, collinearity and noise in data [19,23]. A potential application of PCA during aflatoxin control is coupling it with spectral images to screen signals and identify signals that are specific to aflatoxins [24].

*Support vector machines (SVM)* are algorithms that distinguish between two classes by finding the hyperplane that maximizes the gap between the closest data points of opposite classes. The number of features in the input data dictates if the hyperplane is a line in a 2-D space or a plane in a n-dimensional space. Since multiple hyperplanes may demarcate two classes of data, maximizing the gap between points allows the algorithm to find the best boundary between classes [25]. SVM has diverse applications in disease detection, crop quality classification and yield prediction during agricultural activities.

*Decision trees* are classification or regression algorithms with a tree-like architecture that progressively organize a dataset into smaller homogeneous clusters known as sub-populations. Decision tree algorithms consist of internal nodes that represent a different pairwise comparison of a selected feature. A branch consists of several such nodes and represents the outcome of the comparison [26]. A *random forest* forms when multiple decision trees connect to each other to achieve a single outcome [19]. Random forests are used for crop classification and detection of mycotoxin contamination levels in crops using image analysis [27]. 

*Deep learning* is a specific type of an ANN with complex multilayers that consists of a large number of functions that allow data representation in a hierarchical manner. Deep learning has added advantages of automatic data extraction from raw data, with characteristics from higher levels of the hierarchy being formed by the components of lower levels. Compared to other approaches, deep learning can solve complex problems quickly and effectively due to the complex models used [28]. Similar to PCA, deep learning can be coupled with spectral approaches to detect aflatoxins in crops; however, with better performance [29].

All of these AI methods are useful in solving agricultural and food problems; from robotic harvesting machines operated from a distance, to supply chain solutions in optimal food distribution across multiple sites. Additionally, other technologies can help to ensure food productivity and safety. Many of these fall under modern biotechnologies, discussed next.

## 4. Background: Biotechnology

Biotechnology has its roots in understanding the properties of DNA, genes, and how these can be engineered by humans for specific purposes. The discovery of recombinant DNA and its possibilities in the 1970s has led to breakthroughs in medicine, engineering, and all of the life sciences [30]. Agricultural biotechnology had its first application in 1987 in “Ice-minus”: a genetic variant of the bacterium *Pseudomonas syringae* to spray onto crops to prevent frost damage [31]. Although this product was field-tested, it was never commercialized. Larger agbiotech applications were commercialized and deployed in US agriculture in the mid-1990s: transgenic corn, cotton, potato, and soybean that were modified to produce insecticides (Bacillus thuringiensis (Bt) treated crops), tolerate herbicides (Roundup Ready or Liberty Link crops), or both. To date, a vast majority of the field corn and soybeans produced in the US are biotech varieties. More biotech crops have been developed and commercialized that are not simply transgenic, but gene-edited or genetically modified using RNA-interference methods. Some biotech methods used in agriculture are described below.

*Transgenic crops*, often called genetically modified organisms (GMOs) in the public dialogue, contain genes introduced from foreign (non-host) species using genetic engineering techniques such as gene guns (older method) or *Agrobacterium tumefaciens* to allow gene introgression. These foreign genes confer desired attributes in the host plant such as pest protection, herbicide tolerance, viral resistance, and enhanced nutritional content. As described above, transgenic crops are the earliest biotech crops commercialized worldwide, with the introduction of insect-protected Bt crops and herbicide-tolerant crops in the 1990s in the US and other countries worldwide. Transgenic Bt corn, for example, contains genes from a soil bacterium *Bacillus thuringiensis* (hence Bt) that produce delta-endotoxins, selectively toxic to certain insect pests. Transgenic crops can aid aflatoxin control in several ways. The lower levels of insect pest damage mean fewer entry wounds on kernels that fungi can then easily infect. Additionally, transgenic crops can be developed to express synthetic peptides and enzymes that can retard fungal growth. These are explained in detail in Section 9.

Another type of biotechnology more recently employed in agriculture that can fight against fungal infections is *RNA interference (RNAi)*. RNAi refers to a bioengineering process that allows the host organism (e.g., a corn plant) to interact with another species (e.g., an insect pest or a fungal pathogen) to interfere with particular processes in that species as a protective mechanism to the host. In contrast to Bt corn, which produces proteins that are toxic to insects, RNAi corn silences specific genes in potential pests or pathogens that attack it. In 2017, the US Environmental Protection Agency (EPA) approved a variety of RNAi corn that controls for corn rootworm, to be commercialized in the US (https://www.epa.gov/pesticide-registration/epa-registers-innovative-tool-control-corn-rootworm (accessed on 7 September 2025)). Now, RNAi corn hybrids are being developed that can also control aflatoxin by silencing particular *A. flavus* genes in the aflatoxin biosynthetic pathway [32].

*Clustered regularly interspaced short palindromic repeats (CRISPR)* is an adaptive immune system in bacteria that employs small guide RNAs (crRNAs) to interfere with invading nucleic acids. CRISPR-Cas (also known as CRISPR associated) systems consist of a genomic locus that holds together short repetitive elements (repeats) separated by unique sequences (spacers), which are originally produced in mobile genetic elements (MGEs) such as bacteriophages, transposons, or plasmids. In nature, CRISPR-Cas system acts in a sequence-specific way to recognize and cleave foreign DNA or RNA [33]. In the presence of an invading MGE, a distinct sequence of the MGE is incorporated into the CRISPR array, leading to the formation of a new spacer. The new spacer incorporated CRISPR array is then transcribed into long precursor crRNA and further processed into mature RNA that remembers the invader’s DNA. Finally, the crRNA forms a complex with a Cas nuclease (an enzyme), that ultimately identifies and cleaves the invading DNA with the complementary sequence [33,34]. CRISPR has potential applications in mycotoxin control in food where a guide crRNA could be designed and delivered along with Cas nuclease to cleave genes in fungi responsible for mycotoxin production. In addition to DNA cleavage, different Cas enzymes have different functions that can be utilized for diagnostic purposes. For instance, after recognizing and cleaving a specific target, Cas12a, Cas13, and Cas14 exhibit collateral, nonspecific cleavage activities against single-stranded DNA (ssDNA) or single-stranded RNA (ssRNA). Therefore, by incorporating a ssRNA or ssDNA reporter molecule that can be cleaved by such Cas enzymes resulting in a fluorescent signal, to the CRISPR system, a bio sensor could be designed to generate a signal when a target genetic material is present [35]. Transcriptomics and proteomics can complement CRISPR applications by identifying regulatory targets and resistance traits of plants for reducing the risk of fungal infection and subsequent aflatoxin production.

The remainder of this paper describes how AI and biotechnology can be used to improve aflatoxin detection and control, from seed production to understanding of its toxicology. Figure 1 summarizes how AI could improve methods of aflatoxin control from farm to fork, and beyond to human health.

## 5. AI Applications in Plant Breeding: Conventional or Biotechnological

Today, AI models trained on DNA sequences can generate new sequences encoding potentially novel biological functions. Separately, large language models (LLM) trained on the scientific literature can guide novices in pursuing sophisticated biological work. For example, LLMs can be trained on using the scientific literature about which genes in plants confer resistance to fungal infection and subsequent aflatoxin production. Even more practically, AI can be used in the laboratory to evaluate different crop plant strains with different genetic sequences for *A. flavus* infection, and subsequent aflatoxin contamination, to determine which genes or gene clusters might offer better protection without compromising yield. These genes can then be inserted using conventional breeding or transgenic methods to arrive at crops that are high-yielding and also have lower aflatoxin levels.

Gene expression in plants is affected by environmental stressors, which can be classified into abiotic and biotic stress. Fungal infections are a form of biotic stress that can activate genes responsible for stress responses in the plant. Machine learning has been successfully utilized to identify disease resistance genes specifically under fungal infections in plants including rice [36], sugarcane [37], oilseed rape, and coffee [38]. These studies mainly use genome wide SNP and genotype by sequencing datasets to carry out genomic predictions using different model architectures varying from decision tree, Best Linear Unbiased Prediction (BLUP) models, Recurrent Neural networks to Deep Learning. When these techniques are used, the prediction power of the genomic regions that are responsible for fungal disease resistance was found to be higher compared to traditional methods such as Genome Wide Association Studies (GWAS) [37,39]. Similar approaches can be taken to predict genes responsible for fungal resistance, which then can be used to reduce fungal infections in plants.

## 6. AI Applications in the Field to Detect and Mitigate Aflatoxins

The occurrence of aflatoxins in crops at harvest is impacted by climate and biogeochemical characteristics of the soil that in turn impact both crops and fungi. For instance, soil characteristics such as carbonate content, saturated hydraulic conductivity, pH, bulk density, and texture, influence soil microbial communities, and determine which species of fungi might dominate in a given soil environment [40,41]. Other factors such as fungal dispersion and growth mechanisms also play a role in the occurrence of aflatoxins in crops in a specific soil. Machine learning models can be used to identify the correlation between multiple weather, soil and fungal characteristics, and aflatoxin occurrence to identify risk factors that lead to aflatoxin occurrence in crops [41]. Such knowledge could help develop agricultural practices and mitigation strategies that eliminate or control the identified risk factors.

In a study conducted in corn fields of Illinois, Castano-Duque et al. (2023) [41] employed a gradient boosting machine (GBM) learning models and a neural network (NN) machine learning model to identify the key geospatial and climate properties that correlate with aflatoxin occurrence in corn. GBM identified temperature and precipitation prior to sowing, as significant influential factors contributing to high aflatoxin contamination. The Aflatoxin-GBM model showed that a higher aflatoxin risk index (ARI) in January, March, July, and November, reflecting faster fungal growth and dispersion rates, led to higher aflatoxin contamination in the southern regions of Illinois. Higher values of corn-specific normalized difference vegetation index (NDVI) in July, reflecting denser corn vegetation, led to lower aflatoxin contamination in Central and Southern Illinois. A similar relationship was also observed by Castano-Duque et al. (2022) [42] by using GBM and Bayesian network (BN) modeling.

Computer vision and pattern recognition are AI methods by which we can train models to recognize images corresponding to aflatoxin risk in the field. Right now, very little can help a farmer other than marketing strategies and filing a crop insurance claim [43], if they detect aflatoxin problems in the field; because by the time aflatoxin problems present themselves in the field, it is too late to apply biocontrol or to plant a different hybrid, or even apply good agricultural practices. Currently existing fungicides, which are effective against Fusaria, do not work against Aspergilli. While this is yet to be applied in the field, it could potentially give farmers a head-start on understanding the need for testing, reporting to the US Department of Agriculture, and filing an insurance claim. This will help in funneling potentially contaminated grain to uses that may not require as stringent an aflatoxin standard (e.g., certain types of animal feeds and industrial uses).

Apart from growth conditions and fungal dispersion dynamics that determine the extent of fungal growth in crops, cropping system variables such as sowing and harvesting time and type of tillage determine the extent of contamination that ensues. Using a Deep Neural Network (DNN) approach, Camardo Leggieri et al. (2021) [44] combined both meteorological and cropping system information collected from the fields of Northern Italy to predict the occurrence of aflatoxins in corn with better accuracy compared to traditional linear regression models.

AI could be utilized to identify the antimicrobial activity of peptides, which could potentially be used as a fungicide during cultivation and food processing. Antimicrobial peptides have become more popular in recent decades as an alternative to antibiotics due to their ability to overcome the resistance mechanisms developed by disease pathogens and their ability to generate multiple physiological outcomes by targeting different intracellular components. In particular, their strong DNA binding affinity and the ability to penetrate cell membranes have been identified as key factors for achieving enhanced therapeutic efficacy. Traditional chemoinformatic approaches used to identify antimicrobial activity are less applicable when screening multiple molecules, as exploring large databases becomes a highly tedious and time-consuming task. Instead, trained deep learning models can rapidly screen millions of molecules at once, identifying the most promising candidates within minutes [45]. Although not directly applied to identify the antifungal activity against Aspergilli, recently a deep learning model called AMPs-Net successfully identified four antimicrobial peptides derived from *E. coli*’s fragmented genome. Out of the four identified peptides, three were tested in vitro and all demonstrated antimicrobial activity [45]. Further, detailed peptide—membrane/ligand interactions of such identified molecules could be elucidated using molecular dynamic simulations to predict their biological activity and pharmacological relevance [45,46].

## 7. AI Applications in Food Processing to Reduce Aflatoxins

A key approach to reducing the carry-over of aflatoxins during food processing is sorting the harvest to remove seeds and grains infested with *Aspergillus* species, prior to further processing. Fungal infections generally lead to alterations in the color and appearance of nuts, seeds, and grains, enabling their detection through image processing techniques coupled with AI. In a study conducted with peanuts, Ziyaee et al. (2021) [21] employed three different machine learning tools: ANN, SVM, and adaptive neuro-fuzzy inference system (ANFIS) to detect the presence of *Aspergillus flavus* in images of peanuts seeds taken using LED, UV, and fluorescent lights. The results showed that the combination of LED light in a white background with ANN had 99.7% accuracy in detecting fungal growth on peanuts after 72 h from infection. Similarly, UV lights and a black background with ANFIS achieved 99.9% accuracy in detecting fungal growth after 48 h of infection.

In a similar study, Manhando et al. (2021) [47] employed an error-correcting output code (ECOC)-based SVM trained on features extracted using a pre-trained Deep Convolutional Neural Network (DCNN), to detect *Aspergillus flavus* growth in peanuts using images taken by optical coherence tomography. Their model was successful in identifying fungal growth with a promising accuracy on both the training and testing datasets.

In addition to direct imaging of the infested sample, color changes that take place in colorimetric sensor arrays as a result of mold growth, can also be coupled with AI to detect *Aspergillus* presence in peanuts. For instance, Zhu et al. (2022) [48] used the color changes resulting from the reaction between moldy peanut powder and a colorimetric sensor, to train a support vector regression (SVR) quantitative analysis model that detects AFB1 in peanuts with strong specificity.

Spectral approaches such as fluorescence spectroscopy, and hyperspectral imaging have shown promising results in detecting aflatoxins in food products, due to the fluorescence characteristics of aflatoxins under ultraviolet light [24,49]. However, the multitude of chemical compounds in food matrices can produce autofluorescence, contributing to artifacts in results. Therefore, these approaches require optimization of data acquisition conditions and utilization of suitable chemometric models to analyze results. Machine learning algorithms such as deep neural networks, random forest, SVM, and PCA are previously used alongside spectral approaches to detect aflatoxins in almonds [24], peanuts [27,29,49,50], and figs [51]. Deep learning approaches have shown better performance compared to traditional models such as neural network, random forest and SVM, when detecting aflatoxin contamination using spectral images [29].

In addition to AI-aided detection of aflatoxin for quality control of materials, machine learning could potentially be used to design composite mycotoxin detoxifiers (MDT). MDTs are food or feed additives that form bulky-non absorbable complexes with mycotoxins in the gastrointestinal tract, thereby reducing the absorbability and the bio availability of the toxin. They can also reduce the toxicity of the chemical by promoting their degradation into non-toxic metabolites by the use of bio-transforming agents such as bacteria and enzymes [52]. Development of MDTs on a large scale requires the integration of seemingly conflicting research and development strategies such as computational screening of a large number of materials for their performance and compatibility, high-throughput synthesis and characterization of the material and testing using living animals.

Alternatively, machine learning models can incorporate data obtained from material characterization, mycotoxin assessment and in vitro absorption experiments to identify suitable MDTs for a single or an array of mycotoxins. Using this approach Lo Dico et al. (2022) [52] searched the material space for possible material formulations with high removal capacity of several food toxins including AFB1, while gaining insight into their modes of action. They observed optimum detoxifying performance in a clay composite material consisting of epiolite, montmorillonite, and activated carbon. This builds upon the body of work conducted by Phillips et al. (2019) [53] over decades, on clays that can bind aflatoxin in the intestinal tracts of livestock and humans to allow excretion rather than systemic absorption.

## 8. AI Applications in Understanding Toxicology of Aflatoxins: Whether and How It Affects Child Growth Impairment, Immune System Dysfunction, and Other Diseases and Conditions

It has been known for over 60 years that aflatoxin causes liver cancer, but all the other health effects that have been associated with aflatoxin exposure have a lower weight of evidence, due to a combination of fewer years of research and yet-undetermined mechanisms of toxicity (and thereby unclear associations). Reviewing all the existing literature to evaluate the weight of evidence linking aflatoxin exposure to different health outcomes is a formidable task.

To this end, LLM that effectively and efficiently extracts crucial information from publications on aflatoxin-related health risks, may enable rapid assessment of the weight of evidence linking aflatoxin exposure to the health endpoints in question (e.g., immunological effects, neurological effects, growth stunting). Although LLMs have not yet been used in assessing aflatoxin’s health effects, attempts are currently underway to fine-tune existing Generative Pre-trained Transformer (GPT) models to extract data for chemical risk assessment [54]. While currently available GPT models are increasingly used for general text search and data extraction, model performance decreases as the task becomes more complex and specific. In a case study conducted on bisphenol A (BPA), a GPT3-model (*Curie* base model) was refined using language tokens of 78 publications to answer an array structured questions on the toxicity of the chemical. The fine-tuned model demonstrated markedly better performance than the currently available LLM model used for comparison in the study, suggesting that proper prompt engineering is essential for successful use of LLMs [54].

Complex metabolic pathways that lead to toxin-induced cellular alterations could potentially be modeled using machine learning. One such pathway by which mycotoxins cause cellular damage is lipid peroxidation. For example, in Galvez-Llompart et al. (2023) [22], a machine learning quantitative structure–activity relationship (QSAR) model was employed to predict the lipid peroxidation activity of an array of mycotoxins. QSAR models use molecular descriptors, that is, numerical representations of a molecule’s chemical structure that can capture features such as atom type, shape, and connectivity to predict its ability to induce lipid oxidation. Using an ANN and a linear discriminant analysis approach, the study was able to classify the lipid peroxidation activity of 70 mycotoxins with 88% accuracy.

## 9. Biotechnology Applications in Plant Breeding to Reduce Aflatoxin Occurrence

Transgenic crop technology has enabled the development of crop lines that are resistant to fungal growth, and more recently by directly targeting invading fungi. For instance, Bt corn, which was first developed to confer insect pest control, has significantly less insect damage on its kernels. As kernel wounds allow fungal spores to invade, reduction in insect damage results in reduced mycotoxin occurrence in corn kernels. One of the first commercial applications of transgenic crop technology was the development of Bt corn that contains transgenes from the soil bacterium *Bacillus thuringiensis* that enable the production of crystal proteins, toxic to certain insect pests that reduce corn yield [55]. As kernel wounds caused by insect damage allow fungal spores to invade, reduction in insect damage results in reduced mycotoxin occurrence in corn kernels. Shortly after Bt corn was first commercialized in the US, it was found to have lower *Fusarium verticillioides* infection because of lower insect pest damage, and thereby lower contamination of the mycotoxin fumonisin [56], which has been implicated in neural tube defects and esophageal cancer [57]. More recently, in a study of corn growers’ insurance claims, Bt corn planting was also found to result in lower aflatoxin levels [58]. Newer Bt corn varieties that contain insecticidal proteins in particular, provide significantly improved resistance to the insects such as corn earworm (*Helicoverpa zea*) and fall armyworm (*Spodoptera frugiperda*) that make corn susceptible to aflatoxin accumulation [59]. A recent meta-analysis of the impact of genetically engineered corn suggested that genetically engineered maize results in a 28.8–36.5% reduction in mycotoxin levels [60].

More recent advancements in transgenic corn have been successful in going beyond insect pest control to impart antifungal activity, thereby directly reducing mycotoxin levels in the crop. In particular, transgenic corn expressing the synthetic peptide AGM182 has successfully reduced fungal growth in seeds resulting in a 79–98% reduction in aflatoxin levels [61]. Similarly, transgenic corn transformed using a gene from *Lablab purpureus* beans have successfully expressed α-amylase inhibitor-like protein (AILP) that effectively inhibits α-amylase activity in *A. flavus*, resulting in a 35–72% reduction in *A. flavus* growth [62]. In addition to expressing proteins that can inhibit enzymes necessary for fungal growth, transgenic corn could be developed to express enzymes that chemically degrade aflatoxins present in corn kernels. A recent example is the development of three transgenic maize lines that were modified to express an aflatoxin-degrading enzyme isolated from the edible Honey mushroom *Armillariella tabescens* [63].

Another approach to developing transgenic crops that may inhibit fungal infections is the use of RNA interference (RNAi) in corn. To control aflatoxin occurrence, RNAi corn could enable host-induced gene silencing (HIGS) where the corn plant silences genes in the well-defined aflatoxin biosynthesis pathway of *Aspargillus* [64], allowing the fungi to colonize the corn kernel without producing the toxin [1]. In this approach, the corn plant is modified to produce double-stranded RNA (dsRNA) molecules that correspond to essential genes in the fungal aflatoxin biosynthetic pathway, such as *aflR*, *aflS*, and *omtA*. When *Aspergillus* infects the corn kernel, these dsRNAs are absorbed by the fungus and processed into small interfering RNAs (siRNAs), which guide the degradation of complementary fungal RNAs. This silences the targeted genes, disrupting the expression of enzymes and regulatory proteins necessary for aflatoxin biosynthesis [65].

Maize plants modified with a kernel-specific RNAi gene cassette targeting the *aflC* gene, which encodes an enzyme in the *Aspergillus* aflatoxin biosynthetic pathway, have been successful in stopping aflatoxin occurrence in kernels after pathogen infection [65]. Under similar conditions, transgenic corn with the modified aflC gene did not produce detectable levels of aflatoxin, whereas the non-transgenic control kernels produced toxin levels reaching thousand parts per billion. However, no differences were observed between transcripts from developing aflatoxin-free transgenic kernels and those from non-transgenic kernels [65]. Similarly, a corn variety modified to express an RNAi construct targeting the *A. flavus* alpha-amylase gene *amy1*, has been successful in reducing *amy1* gene expression thereby reducing fungal colonization and aflatoxin accumulation in kernels [66]. Development of a maize variety incorporated with an RNAi vector containing a portion of the *A. flavus* Alkaline protease (alk) gene has resulted in an 87% reduction in both *A. flavus* growth and aflatoxin accumulation under laboratory conditions [32]. In field conditions, corn incorporated with an RNAi vector containing a portion of the *aflM* gene that targets versicolorin dehydrogenase, a key enzyme involved in the aflatoxin biosynthetic pathway, reduced aflatoxin presence in kernels by up to 60% compared to the non-transgenic samples when planted in the field and allowed to self-pollinated by hand [67].

As opposed to transgenic crops, which are produced by artificially inserting an exogenous DNA stretch into a plant genome, usually from a foreign species to achieve a desired trait, CRISPR can achieve desired attributes by precisely editing the host’s DNA itself. While CRISPR has not been used directly for aflatoxin reduction in crops, recent advances in this technology may prove useful to reduce aflatoxins in crops. For instance, a CRISPR/Cas system was successfully used to delete large genomic fragments in polykaryotic fungi that blocked the production of the mycotoxin citrinin, produced by the fungus species *Monascus purpureus*, which contaminates Monascus Red food pigment [68]. In vitro reconstituted CRISPR-Cas9 has successfully been used to induce targeted gene deletions in *Penicillium polonicum*, that disabled production of verrucosidin, a mycotoxin commonly found in food such as fresh and cured meat, nuts, and some fruits [69].

These examples demonstrate the potential of using CRISPR technology to lower the risk of aflatoxin contamination in corn by targeting and disrupting the biosynthetic pathways responsible for toxins formation, or by modifying host traits that make corn susceptible to fungal infection. These traits may involve genes related to heat tolerance, drought resistance, and defense against insect pests. Although such characteristics are often controlled by multiple genes, advances in multiplex CRISPR and other gene editing tools may soon enable effective modifications to achieve desired phenotypic traits [1].

Machine learning and genomic prediction models could potentially be coupled with biotechnology to assist researchers in identifying genomic regions associated with aflatoxin resistance in crops. By analyzing large datasets of genetic markers and phenotypic traits, these models detect patterns and estimate the contribution of specific loci to resistance. The identified genomic regions can then be cross-referenced with GWAS or functional annotations to pinpoint candidate genes for further modification or incorporation into resistance breeding programs.

## 10. Biotechnology Applications to Enhance Aflatoxin Detection in Food and Feed

CRISPR is increasingly used to develop biosensors that detect aflatoxins in food. Since aflatoxins are small, non-nucleic acid molecules, CRISPR cannot detect them directly. However, biosensors may be developed via indirect strategies that convert aflatoxins into a specific DNA-based signal identifiable by CRISPR technology.

One indirect strategy is using an aptamer—activator complex; a composite molecule made by coupling an aptamer (a synthetic DNA or RNA molecule with a specific 3D structure that binds to a target analyte) to an activator protein. In the presence of aflatoxin, the aptamer binds to it, while the activator triggers the activation of the CRISPR-Cas system. Depending on the design, the CRISPR-Cas system performs its diagnostic function, which is typically the collateral cleavage of a reporter molecule (generally a fluorescently labeled ssDNA or ssRNA), which generate a signal upon cleavage. This approach was successfully used by Zhu and Zhao (2024) [70] who developed a dual functional DNA probe by linking an AFB1 aptamer with Cas12a activator. They prepared the diagnostic probe by linking the AFB1 aptamer—A32 and the acDNA of Cas12a (activator) via a poly-thymine (poly-T) space sequence (A32-acDNA). A reporter system was separately prepared as a solution containing a ribonucleoprotein complex (RNP) and fluorescently labeled T10 reporter. For detection, a sample containing AFB1 should be mixed with the probe to facilitate the preferential binding between the probe and AFB1. Then the binding solution is applied on a microplate coated with bovine serum albumin (BSA)-AFB1 conjugates, where free unbound probes are captured on the microplate by the high binding affinity between A32 aptamers and AFB1 conjugates on the microplate. The microplate containing the attached probes is then incubated in the reporter solution, where acDNA region interacts with RNP complex, leading to the cleavage of the fluorescently labeled reporters and the production of fluorescence signal. With the increase in AFB1 in the tested sample, more A32-acDNA probes binds to the free AFB1, reducing the quantities captured by BSA-AFB1 on the microplate. This results in a concentration dependent reduction in the fluorescent signal when immersed in the reporter solution, facilitating AFB1 quantification [70].

Another, novel fluorescence biosensor was constructed with CRISPR/Cas12 by intergrading MXenes-Ti_3_C_2_Tx, a novel two-dimensional metal carbide with a surface rich in –OH, –F, and –O groups, which can easily interact with biomolecules in the CRISPR/Cas12 system. The biosensor had a detection range from 0.001 to 80 ng mL^−1^, with a detection limit of 0.92 pg mL^−1^, and the ability to detect within 80 min. It demonstrated excellent detection performance in real peanut samples [71]. Similarly, a multimodal sensor with detection limits of 0.85, 0.79, and 1.65 pg mL^−1^ was constructed by a CRISPR/Cas12a system consisting of tetramethylbenzidine as a colorimetric agent for visual detection of AFB1 [72].

Nano particle technology often is coupled with CRISPR-Cas systems to enhance the sensitivity of the biosensors. In a recent study, activator strands were introduced to initiate the *trans*-cleavage of CRISPR/Cas12a for cutting the nanoparticles-tagged-magnetic beads, which were transduced to nanoparticle count signals measurable by single-particle mode inductively coupled plasma mass spectrometry (Sp-ICPMS) [73]. In another study [74], a surface-enhanced Ramen Spectroscopy (SERS) platform was developed by combining CRISPR/Cas12 and nano particle technology. SERS is increasingly used for diagnostics purposes; however, the beacon molecules used for SERS detection often overlap with the signal of biological samples, disturbing the target signal. In an attempt to resolve this problem, Zhang et al. (2023) [74] synthesized a sensor consisting of nano particles embedded with the dye molecule, Prussian blue, which is released and gives a signal in the Ramen silent region, enabling interference free detection when coupled with the CRISPR/Cas 12 system. Visual detection of AFB1 was accomplished by Chen et al. (2023) [75] who used multiple isothermal amplifications coupled with CRISPR/Cas14a. This technique utilized composite nanoprobes comprising magnetic nanoparticles and gold nanoparticles. AFB1 was efficiently identified through an aptamer competition process where the activation of CRISPR/Cas14a in the presence of AFB1 triggers the cleavage of composite nano probes and nano particle release, resulting in significant changes in both color and colorimetric signal.

## 11. Case Study in AI and Biotechnology to Reduce Aflatoxin from Farm to Fork

The previous discussion demonstrates how one or a number of combined potential strategies could be deployed—with a consideration for cost—to reduce aflatoxin risk at different points in the food supply chain. Below is a hypothetical scenario of how a suite of AI- and biotechnology-informed strategies, could be applied to corn, a key crop frequently contaminated with aflatoxin worldwide.

First, the development of corn varieties that are more resistant to *A. flavus* infection and subsequent aflatoxin production without compromising yield may benefit from high-throughput screening of genes and gene clusters not just in existing corn varieties, but in other species that provide these desired qualities. These may aid in breeding—through either conventional or transgenic methods—corn varieties that are better able to resist aflatoxin while still providing high yields. To date, it has been a challenge to find fungal-resistant varieties of crops that do not also suffer from compromised yield. AI may aid crop breeders in finding the right gene or combination of genes across multiple existing landraces (traditional varieties) and hybrids of corn that have both acceptable yield performance and a greater ability to resist *A. flavus* infection and subsequent aflatoxin production.

Additionally, biotech corn varieties, such as described above, can resist insect damage that predisposes the corn to fungal infection (transgenic Bt corn) or in the future, can degrade aflatoxin in the kernels to reduce human and animal exposures.

Machine learning may be applied to historical data of both climatic conditions (daily temperatures, relative humidity, precipitation, and soil moistures) and real-time images of corn throughout growth stages, to identify and be trained on conditions that result in high aflatoxin risk. For example, there might be a visual cue that human eyes have missed or cannot easily discern in a large corn field (discoloration, spotting, etc.), which indicates that an aflatoxin problem may occur; (i.e., detected by computer vision and pattern recognition enabled by machine learning, that is not evident to the human eye). Then new biotechnological methods of developing peptides that are specifically antifungal to *Aspergillus* could be applied to the corn (unlike biocontrol, it could be applied early in fungal infection and not at the very start of the growing season).

In postharvest conditions, AI combined with biotechnology could be used to reduce aflatoxin risk. Currently, when harvested corn is taken to a grain elevator, aflatoxin measurements are frequently inaccurate. Aflatoxin can concentrate in a very small segment of an entire consignment: in one or a few kernels, or in a “hot spot” in a bin [76]. This is the main source of sampling error: what has traditionally made sampling for aflatoxin analysis difficult. If the elevator operator happens to take an analytical sample that either has a much higher or lower aflatoxin concentration than the rest of the lot on average, it is an inaccurate measure of the actual aflatoxin in that bin. One method of dealing with this currently is for the farmer to request an appeal sample if the analytical sample showed excessively high aflatoxin levels for market purposes [77]; but overall, this practice biases analyses to lower aflatoxin levels than that may exist. A combination of AI and biotechnology, through computer vision and heat maps for aflatoxin detection, could allow farmers to find the hot spots of aflatoxin in their bins; allowing the them to remove or separate the contaminated portion for destruction or for sale in a market that allows higher aflatoxin levels (e.g., ethanol or other industrial production).

If, despite all control efforts, aflatoxin nonetheless finds its way into human diets, more solutions may be possible using biotechnologies to reduce risks to human health. As described earlier, certain compounds in *Brassica* and *Allium* vegetables are bio-transformed in human digestion to induce Phase 2 enzymes that help detoxify aflatoxin. Conceivably, these compounds or their metabolites (sulforaphane, allicin) could be engineered into corn so that the corn could produce these compounds as well. Then even if aflatoxin is present, its harmful effects could be partially ameliorated in the diet, if *Brassica* and *Allium* vegetables are not easily available.

## 12. Discussion

It is an exciting time for new technologies, particularly as they converge and become more available and accessible to diverse stakeholder groups, including the general public. There is no reason that these new technologies could not be used in agriculture to solve global, millennia-old societal problems, including those surrounding foodborne mycotoxins. In particular, aflatoxin—the most widely recognized mycotoxin because of its toxicity, carcinogenicity, and prevalence in foods worldwide—has continued to cause economic, and human and animal health risks in some of the most vulnerable parts of the world. While a number of control strategies have been developed, the implementation and success of these has varied widely worldwide.

Now AI and biotechnology have both matured to a level that allows not just companies, but the general public, to access them to a higher degree than ever before. In this paper, we have described how these technologies can be used from farm to fork to reduce aflatoxin risk, to a degree that is likely to substantially reduce it so that human and animal health can be protected from its serious effects. In particular, what these technologies offer is a level of precision and accuracy in relatively quick time that previously would have taken an extremely long time, and large amounts of resources, to achieve. The benefit is that solutions emerging from these technologies could be deployed anywhere in the world.

Nonetheless, barring a few field trials [41,44] that have utilized AI to identify geospatial, climatic, and cropping system variables contributing to aflatoxin occurrence, most AI technologies currently under investigation have yet to be implemented on a large scale in real-world settings. Even the studies reviewed in this work are largely concentrated in developed countries such as those in Europe and the United States. Research efforts in developing regions—including Asia, Africa, and Latin America, where aflatoxin prevalence is particularly high—remain scarce. Significant progress is still needed in customizing and training LLMs which is still at the preliminary stage of being used in toxicology, to extract and analyze data from the existing literature to predict metabolic pathways through which aflatoxins induce cellular damage, assess toxicity, and perform automated risk analyses to identify diseases associated with aflatoxin exposure.

One of the primary challenges limiting AI application in aflatoxin control is the need for extensive, high-quality, and well-labeled datasets to train reliable models. Acquiring such datasets requires substantial human expertise and financial resources, both of which are often limited in developing nations. AI for aflatoxin detection may falter in the field due to poor cross-region transfer (different climates, cultivars, storage, and *Aspergillus* species) and sensor drift or inconsistent calibration that distorts the results. Additionally, the high initial investment required to deploy AI-based technologies such as computer vision systems and spectrometric approaches in agricultural fields serves as a deterrent for many traditional farmers.

Similarly, biotechnology-based tools such as CRISPR-enabled biosensors remain at the bench-scale stage and involve several labor-intensive steps before results can be obtained. Considerable research is still required to scale up these technologies and develop user-friendly, in situ devices suitable for diverse farming and food processing environments.

Beyond scaling challenges, we identify two short-term research directions that could advance the integration of AI and biotechnology for aflatoxin control. The first is a quantitative comparison of the accuracy, reproducibility of computer vision and machine learning–assisted detection approaches against conventional detection techniques such as ELISA and liquid chromatography–tandem mass spectrometry (LC-MS/MS). Such a study coupled with a cost benefit analysis would assess the feasibility of employing these technologies on a large scale. The second is a quantitative feasibility assessment of the proposed case study of integrating of AI and biotechnology across the corn supply chain, using data from both the existing literature and future field trials.

These potential solutions and directions described above, however, are highly dependent on nations’ and communities’ acceptance of AI and biotechnologies. Over the last decade, several nations that have had recorded problems of aflatoxin have commercialized transgenic Bt corn that can help to reduce this problem, such as Kenya (https://www.isaaa.org/gmapprovaldatabase/ (accessed on 7 September 2025)). But notably, Bt corn has already been commercialized in other nations for nearly 30 years. It is likely that any further biotech crop application approvals would take considerably longer. By contrast, AI regulation at a national level has been relatively lacking worldwide. Despite legitimate ethical and moral considerations, which regulators and the industry are actively managing, AI and biotechnology present a timely avenue to mitigate aflatoxin contamination and associated public-health risks. We encourage further research in these areas to combat the millennia-old problem of aflatoxin, to ensure a safer food supply worldwide.

## Figures and Tables

**Figure 1 toxins-17-00524-f001:**
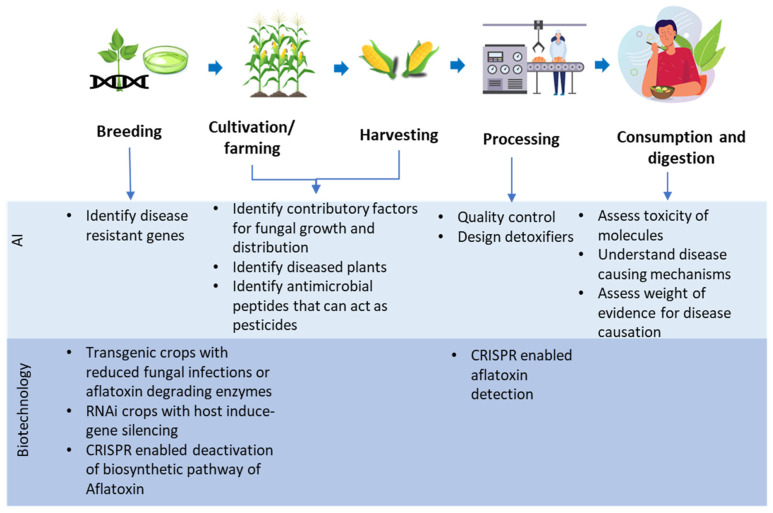
Potential areas of aflatoxin control that AI and biotechnology can enhance.

## Data Availability

The original contributions presented in this study are included in the article. Further inquiries can be directed to the corresponding author.

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
