# Peer review of "AI and Biotechnology to Combat Aflatoxins: Future Directions for Modern Technologies in Reducing Aflatoxin Risk"

_toxins, 2025, doi:10.3390/toxins17110524_

Round 1
Reviewer 1 Report
Comments and Suggestions for Authors
The topic of the manuscript is interesting and worthy of publication. The importance of mycotoxins in food safety and the constant risk they pose justify the importance of the topic. The applicability of artificial intelligence and biotechnological innovations to reduce risks is a very promising area. The manuscript summarizes this. It may be useful for professionals working in this field. Well-written manuscript. Apart from the excessive spacing in lines 450-455, there are no formal deficiencies. I found no professional objections to the manuscript. It can be published in its current form.
Author Response
Comment 1: The topic of the manuscript is interesting and worthy of publication. The importance of mycotoxins in food safety and the constant risk they pose justify the importance of the topic. The applicability of artificial intelligence and biotechnological innovations to reduce risks is a very promising area. The manuscript summarizes this. It may be useful for professionals working in this field. Well-written manuscript. Apart from the excessive spacing in lines 450-455, there are no formal deficiencies. I found no professional objections to the manuscript. It can be published in its current form.
Response 1: We thank the reviewer for the encouraging feedback. We have revised the manuscript to remove excessive spacing in lines 450-455.
Reviewer 2 Report
Comments and Suggestions for Authors
The manuscript sets out to explore how artificial intelligence (AI) and modern biotechnologies can be harnessed for aflatoxin-related research, detection, and mitigation across the entire food chain—from agricultural production to human health. This integrative aim is commendable and reflects a systems-level understanding of aflatoxin risk, aligning well with interdisciplinary trends in food safety and public health. However, the authors should clarify whether the article intends to propose a roadmap for future research or primarily synthesize existing applications. Emphasizing the novelty and unique contribution of their integrative perspective—especially in contrast to previous reviews—would strengthen the manuscript’s positioning.
Section 2. This section is currently too brief and lacks the depth. It should be expanded to include: toxicodynamics and biosynthetic pathways (e.g., aflatoxin B1 metabolism), global prevalence and burden of aflatoxin exposure, regulatory frameworks (e.g., EU vs. Codex standards)
Section 3. While informative, this section is relatively elaborate and includes general definitions and applications beyond aflatoxin control. Risks shifting the article’s focus away from its core toxicological theme. The authors should concentrate on AI tools directly relevant to aflatoxin detection, prediction, and mitigation.
Section 4. The section would benefit from a clearer focus on actionable biotechnological strategies tailored to aflatoxin control: emphasize biosensors, genetic engineering of crops for resistance, and omics-based approaches for pathway disruption. Authors should integrate transcriptomics and proteomics to complement CRISPR applications, particularly in identifying regulatory targets and resistance traits.
Section 6. This section is well-positioned but could be strengthened by including validated field trials or pilot programs to demonstrate real-world applicability. Authors should discuss practical barriers such as data availability, regional generalizability, and sensor calibration. A comparison between AI-based detection methods and conventional techniques (e.g., ELISA, HPLC) will add more value
Section 9. Authors should include a brief mention of how machine learning or genomic prediction models assist in identifying resistance loci and a more detailed explanation of how biotechnological interventions disrupt fungal colonization or aflatoxin biosynthesis (e.g., via antifungal proteins, RNA interference)
Section 10 The section should be expanded to include examples of CRISPR-based genome editing targeting aflatoxin biosynthetic genes in Aspergillus species and integration of omics approaches (e.g., transcriptomics, proteomics) to identify expression markers and guide biosensor design
Section 11 The case study is a valuable addition but would benefit from inclusion of quantitative data or modeled outcomes (e.g., reduction in aflatoxin exposure, cost-benefit analysis) and clarification on whether the case study is hypothetical, based on pilot data, or drawn from published field trials
Author Response
Comment 1: The manuscript sets out to explore how artificial intelligence (AI) and modern biotechnologies can be harnessed for aflatoxin-related research, detection, and mitigation across the entire food chain—from agricultural production to human health. This integrative aim is commendable and reflects a systems-level understanding of aflatoxin risk, aligning well with interdisciplinary trends in food safety and public health. However, the authors should clarify whether the article intends to propose a roadmap for future research or primarily synthesize existing applications. Emphasizing the novelty and unique contribution of their integrative perspective—especially in contrast to previous reviews—would strengthen the manuscript’s positioning.
Response 1: We thank the reviewer for their constructive feedback. In this paper, we have synthesized findings from prior literature and presented a hypothetical scenario illustrating how AI and biotechnology could be fully integrated into the entire food supply chain of corn. The last paragraph of the introduction (lines 47–57) has been revised to more clearly highlight the uniqueness and novelty of this study.
Comment 2. Section 2 - This section is currently too brief and lacks the depth. It should be expanded to include: toxicodynamics and biosynthetic pathways (e.g., aflatoxin B1 metabolism), global prevalence and burden of aflatoxin exposure, regulatory frameworks (e.g., EU vs. Codex standards)
Response 2: A detailed description of how AFB1 is metabolized in the liver was added in lines 76-83. Another section explaining the prevalence and global burden of liver cancer attributed to AFB1 exposure was added in lines 87- 94. Due to the large number of varied regulatory limits practiced globally for aflatoxin control, they are not presented in the current paper keep the emphasis on AI and biotechnologies. Instead, readers are directed to Wu et al. 2013 in lines 94-98.
Comment 3 - Section 3: While informative, this section is relatively elaborate and includes general definitions and applications beyond aflatoxin control. Risks shifting the article’s focus away from its core toxicological theme. The authors should concentrate on AI tools directly relevant to aflatoxin detection, prediction, and mitigation.
Response 3: We had written this section in great detail to explain basics of AI and ML to non-experts. However, upon reviewer’s comments we have shortened the second paragraph. We have also revised lines 139-142, 150-152, 166-167 and 173-175 to highlight how each ML algorithm can be used for aflatoxin control.
Comment 4 -Section 4. The section would benefit from a clearer focus on actionable biotechnological strategies tailored to aflatoxin control: emphasize biosensors, genetic engineering of crops for resistance, and omics-based approaches for pathway disruption. Authors should integrate transcriptomics and proteomics to complement CRISPR applications, particularly in identifying regulatory targets and resistance traits.
Response 4: We have revised lines 206-209 to better explain the direct applications of transgenic crops for fungal resistance. The applications of RNAi and CRISPR in developing GMOs and biosensors were described in the original version in lines 219-220 and 235-244 respectively. We have also discussed the roles of transcriptomics and proteomics to complement CRISPR applications in identifying regulatory targets and resistance traits.
Comment 5 - Section 6. This section is well-positioned but could be strengthened by including validated field trials or pilot programs to demonstrate real-world applicability. Authors should discuss practical barriers such as data availability, regional generalizability, and sensor calibration. A comparison between AI-based detection methods and conventional techniques (e.g., ELISA, HPLC) will add more value.
Response 5: We thank the reviewer for the suggestion. While filed trials are very limited, we in fact included 2 field trials (Castano-Duque et al., 2023; Camardo Leggieri et al. 2021) in this section, how ever we had not specified Camardo Leggieri et al. (2021) as a field trial. We have now revised line 314 to convey that this was a field trial. We were not able to find field trails that applied computer vision for the detection of aflatoxin detection. We therefore propose it as a potential application in line 305. We agree with the reviewer that a comparison of detection technologies would be useful, and propose it as a future direction lines 650-653. Practical barriers for implementing AI in the field rediscussed in lines 667-674.
Comment 6 Section 9. Authors should include a brief mention of how machine learning or genomic prediction models assist in identifying resistance loci and a more detailed explanation of how biotechnological interventions disrupt fungal colonization or aflatoxin biosynthesis (e.g., via antifungal proteins, RNA interference)
Response: We have provided a brief description of machine learning or genomic prediction models assist in identifying resistance loci in lines 515-521. Lines 470-476 now describes in detail, the mechanism by which RNAi disrupt fungal colonization.
Comment 7- Section 10 The section should be expanded to include examples of CRISPR-based genome editing targeting aflatoxin biosynthetic genes in Aspergillus species and integration of omics approaches (e.g., transcriptomics, proteomics) to identify expression markers and guide biosensor design
Response 7: The comment is not clear to us. Section 10 focuses on the applications of biotechnology for detecting aflatoxin, not on the use of genome editing tools such as CRISPR to inhibit aflatoxin production. In fact, we discuss the potential applications and current limitations of employing CRISPR-based gene editing to reduce aflatoxin occurrence in crops in lines 496–506. To the best of our knowledge, CRISPR assisted biosensors for aflatoxin detection are developed by using an aptamer – activator complex. This method is now described in more detail with an example in lines 527-550. We are happy to revise this section again after further clarification from the reviewer.
Comment 8- Section 11 The case study is a valuable addition but would benefit from inclusion of quantitative data or modeled outcomes (e.g., reduction in aflatoxin exposure, cost-benefit analysis) and clarification on whether the case study is hypothetical, based on pilot data, or drawn from published field trials
Response 8: While we agree with the reviewer’s suggestion, conducting a quantitative analysis is out of the scope of the current study. The focus herein is to propose a hypothetical scenario (clarified in line 585) where all introduced AI and biotechnology tools could be integrated to the corn food supply chain to enable better mycotoxin control. Nonetheless we propose a quantitative analysis as a possible future direction in lines 687-689.
Reviewer 3 Report
Comments and Suggestions for Authors
This paper focuses on the key issue in the field of food safety, which is the synergy between AI and biotechnology to combat aflatoxins. The topic has both academic value and practical significance. As a global food safety hazard that has been unresolved for a century, the prevention and control of aflatoxin requires integration and innovation across technical fields. The paper, based on the "Farm to Fork" full chain framework, systematically elaborates on the integration and application of AI and biotechnology, filling the limitations of a single technical perspective.
- There are insufficient cases of deep integration between AI and biotechnology, and most of the links are still in the preliminary stage: although the paper emphasizes "collaboration", there is relatively little discussion and proof of some integration scenarios; The application of LLM in toxicology research: Only through the analogy of Sonnenburg's (2024) study on bisphenol A, there is a lack of direct application cases in the field of aflatoxin (such as LLM extracting literature data on the association between AFB1 and child growth and development), resulting in a lack of targeted support for the conclusion of "AI assisted toxicology assessment".
- Insufficient discussion on practical application challenges and limitations: Lack of connection between laboratory and industrial scenarios: AI image detection (such as 99.9% accuracy) and CRISPR sensors (80 minute detection) in food processing are both results obtained under laboratory conditions, without mentioning actual industrial scenarios such as interference factors during large-scale sample processing and equipment cost limitations for small and medium-sized enterprises
- Illustration and chart suggestions: It is recommended to add a schematic diagram to show the intervention points of AI and biotechnology in the entire chain from farm to table. Consider adding tables to summarize the application effects of AI models in different scenarios such as breeding, testing, and toxicology.
-
In Paragraph 6, two models are mentioned, yet there is no detailed explanation of their evaluation metrics such as prediction accuracy and recall rate. Is it necessary to mention the scale and source of the training data to ensure the rigor of the data source?
- Most of the cases and studies in the article focus on Europe and the United States, while discussions on the technical adaptability in Asia, Africa, and Latin America—regions with high aflatoxin prevalence—are extremely limited. Is it necessary to supplement cases from Asia, Africa, and Latin America to ensure the rigor of the article?
- The conclusion only states: "We encourage research and development in these areas to improve the precision, accuracy, and speed in addressing this food safety risk." Is it necessary to specify the core directions for future research to provide guidance for readers?
- Insuffcient discussion on technical limitations:The paper emphasizes the advantages of Aland biotechnology but mentions little aboutthe practical obstacles to theirimplementation.For instance, Al models rely ona large amount of high-quality labeled databut data collection capabilities are weak intropical and subtropica underdevelopedregions(high-risk areas for afatoxin).
Author Response
Comment 1: This paper focuses on the key issue in the field of food safety, which is the synergy between AI and biotechnology to combat aflatoxins. The topic has both academic value and practical significance. As a global food safety hazard that has been unresolved for a century, the prevention and control of aflatoxin requires integration and innovation across technical fields. The paper, based on the "Farm to Fork" full chain framework, systematically elaborates on the integration and application of AI and biotechnology, filling the limitations of a single technical perspective.
Response 1: We thank the reviewer for the careful appraisal of our work and the constructive feedback. We note that majority of the concerns raised by the reviewer are yet to be investigated in detail, and are real limitations when it comes to deploying AI and biotechnology in the field. Therefore, we have included a separate discussion of the limitations in lines 655-679.
Comment 2: There are insufficient cases of deep integration between AI and biotechnology, and most of the links are still in the preliminary stage: although the paper emphasizes "collaboration", there is relatively little discussion and proof of some integration scenarios; The application of LLM in toxicology research: Only through the analogy of Sonnenburg's (2024) study on bisphenol A, there is a lack of direct application cases in the field of aflatoxin (such as LLM extracting literature data on the association between AFB1 and child growth and development), resulting in a lack of targeted support for the conclusion of "AI assisted toxicology assessment".
Response 2: We agree with the reviewer and discuss the need for customizing and training existing LLMs to be effectively used in toxicology in lines 662-665.
Comment 3: Insufficient discussion on practical application challenges and limitations: Lack of connection between laboratory and industrial scenarios: AI image detection (such as 99.9% accuracy) and CRISPR sensors (80 minute detection) in food processing are both results obtained under laboratory conditions, without mentioning actual industrial scenarios such as interference factors during large-scale sample processing and equipment cost limitations for small and medium-sized enterprises.
Response 3: These aspects are now discussed in lines 676-679.
Comment 4: Illustration and chart suggestions: It is recommended to add a schematic diagram to show the intervention points of AI and biotechnology in the entire chain from farm to table. Consider adding tables to summarize the application effects of AI models in different scenarios such as breeding, testing, and toxicology.
Response 4: We in fact included a schematic representation as figure 1 in our original submission. It is now replaced with a better formatted version. The current paper follows a format where each AI application is first introduced and applications are summarized sequentially afterwards. We feel including a table with summary details would be a repetition of information.
Comment 5: In Paragraph 6, two models are mentioned, yet there is no detailed explanation of their evaluation metrics such as prediction accuracy and recall rate. Is it necessary to mention the scale and source of the training data to ensure the rigor of the data source?
Response 5: We are not sure what the reviewer meant here. Both paragraph s6 summarizes the introduction and seven provides and introduction to AI. None of them seem to contain information about specific models.
Comment 6: Most of the cases and studies in the article focus on Europe and the United States, while discussions on the technical adaptability in Asia, Africa, and Latin America—regions with high aflatoxin prevalence—are extremely limited. Is it necessary to supplement cases from Asia, Africa, and Latin America to ensure the rigor of the article?
Response 6: We found that applications of AI and biotechnology for mycotoxin are very limited in developing countries. We therefore discuss this observation along with the reasons provided by the reviewer in the last comment in lines 658-662 and in lines 667-670.
Comment 7: The conclusion only states: "We encourage research and development in these areas to improve the precision, accuracy, and speed in addressing this food safety risk." Is it necessary to specify the core directions for future research to provide guidance for readers?
Response 7: We agree. We propose two core directions in lines 681-688 along with scaling up effortsas potential areas of future work.
Comment 8: Insufficient discussion on technical limitations: The paper emphasizes the advantages of Aland biotechnology but mentions little about the practical obstacles to their implementation. For instance, Al models rely on a large amount of high-quality labeled data but data collection capabilities are weak in tropical and subtropical underdeveloped regions (high-risk areas for aflatoxin).
Response 8: As described in our response to comment 1, we have now included a separate section on technical limitations in lines 655-680.
Round 2
Reviewer 3 Report
Comments and Suggestions for Authors
The author has made the necessary modifications as requested and has no other suggestions